# Deciphering the Tumor Microenvironment in Prostate Cancer: A Focus on the Stromal Component

**DOI:** 10.3390/cancers16213685

**Published:** 2024-10-31

**Authors:** Hubert Pakula, Filippo Pederzoli, Giuseppe Nicolò Fanelli, Pier Vitale Nuzzo, Silvia Rodrigues, Massimo Loda

**Affiliations:** 1Department of Pathology and Laboratory Medicine, Weill Cornell Medicine, New York, NY 10021, USA; hup4002@med.cornell.edu (H.P.); fip4001@med.cornell.edu (F.P.); gif4002@med.cornell.edu (G.N.F.); pvn4001@med.cornell.edu (P.V.N.); sdr4001@med.cornell.edu (S.R.); 2Sandra and Edward Meyer Cancer Center, Weill Cornell Medicine, Belfer Research Building, 413 East 69th Street, New York, NY 10021, USA; 3Department of Oncologic Pathology, Dana-Farber Cancer Institute and Harvard Medical School, 450 Brookline Ave, Boston, MA 02215, USA; 4Nuffield Department of Surgical Sciences, University of Oxford, Oxford OX1 2JD, UK

**Keywords:** tumor microenvironment (TME), cancer-associated fibroblasts (CAFs), mesenchymal cells, prostate cancer (PCa), castration-resistant prostate cancer (CRPC)

## Abstract

Prostate cancer is one of the most common cancers in men, and its progression is strongly influenced by the surrounding environment, known as the tumor microenvironment. This is comprised of a variety of cells, including cancer-associated fibroblasts (CAFs), that modulate tumor initiation and progression and may affect treatment strategies. This review focuses on the interactions between fibroblasts and epithelial cancer cells and how these interactions contribute to cancer progression. This knowledge may unveil novel therapeutic approaches that could mitigate aggressiveness and/or render prostate cancer more responsive to current treatments, thus improving patient outcomes.

## 1. Introduction

Prostate cancer (PCa) is the second most prevalent cancer in men globally [1]. Most PCas are adenocarcinomas, which arise from the epithelial cells of the prostate gland [2,3]. Androgens are crucial in the development and maintenance of the prostate and drive PCa. Thus, interference with androgen receptor (AR) signaling is the mainstay of therapy for this disease [4,5,6]. Gonadal androgen deprivation therapy (ADT) [7,8,9] and androgen receptor signaling inhibitors (ARsi) eventually result in resistance, leading to the emergence of castration-resistant prostate cancer (CRPC) [10,11,12]. Importantly, the tumor microenvironment (TME), which includes both immune and non-immune cells, also plays a crucial role in the initiation, maintenance, and progression of PCa [13,14]. The TME consists of various types of cells, including mesenchymal/stromal cells, endothelial cells, fibroblasts, myofibroblasts, and immune cells [14,15,16,17]. These cells release chemokines, cytokines, extracellular matrices, and matrix-degrading enzymes and interact with benign and malignant epithelial cells. Moreover, cancer and stromal cells are subjected to mechanical and selective pressures that help them survive carcinogens, hypoxia, inflammation, and chemotherapeutic drugs. These adaptive changes affect disease progression, tumor metastasis, and resistance to conventional therapy [18]. This review will examine novel subtypes of prostate cancer-associated stromal cells, the relationship and crosstalk between epithelial and these mesenchymal cells, and how this relationship affects tumor progression and can be exploited therapeutically. Genetic alterations that induce mesenchymal states and related biomarkers are described. 

## 2. Mesenchymal Cells in the Prostate Tumor Microenvironment

Mesenchymal cells are a heterogeneous collection of cells that arise from the mesodermal germ layers during embryonic development [19,20]. Putative mesenchymal stem cells (MSCs) likely differentiate into stromal cell types, such as fibroblasts, smooth muscle cells, pericytes, adipocytes, osteoblasts, and chondroblasts [21,22]. While MSCs are primarily located in the bone marrow, they can also be found in other tissues, including the prostate. MSCs can migrate and efficiently impede tumor progression, for instance, via microRNA exchange, exosome secretion, or through stimulation of angiogenesis [23,24]. Stromal cells can also suppress the immune system [25]. Prostate stroma is derived from the embryonic urogenital sinus mesenchyme (UGSM) [26]. The activation of the androgen receptor (AR) in the urogenital sinus mesenchyme (UGSM) is essential for the growth and specialization of the epithelial cells [27]. In turn, epithelial cells promote the growth of smooth muscle cells and fibroblasts [28]. The proximity of mesenchymal cells to epithelial cells is crucial for establishing and preserving organ homeostasis (Figure 1A,B).

In adult human and mouse prostates, smooth muscle cells express smooth muscle cell markers such as actin (αSMA) [15]. Other markers, such as fibroblast activation protein (FAP), platelet-derived growth factor receptor-α (PDGFRα), and the intermediate filament vimentin (VIM), can be utilized for identification purposes [30], although their expression is not entirely specific. Single-cell sequencing (scRNA-seq) has allowed the in-depth molecular analysis of stromal populations and an understanding of the diversity within the prostatic stroma beyond what can be recognized by morphology alone. Karthaus et al. performed an in-depth analysis of mesenchymal cells in the wild-type mouse prostate using scRNA-seq, which uncovered a higher level of complexity than previously believed [16]. Two distinct mesenchymal subgroups (Mes1 and Mes2), as well as smooth muscle cells, were identified [16]. Mes1 cells exhibited the expression of *Wnt2*, *Wnt6*, *Wnt10a*, and *RorB*, whereas Mes2 cells had the expression of *Rspo1*, *Fgf10*, and *Sult1e1* [16]. The identification of smooth muscle cells was based on the expression of crucial contractile genes, including *Acta2* and *MyH11*. Of note, a variable percentage of all these stromal cells express the *Ar* [16]. Kwon et al. identified three different stromal clusters (R1, R2, and R3) using sequencing, flow cytometry, and immunostaining in WT mice with distinct transcriptional patterns and likely functions [15]. The R1 subpopulation, characterized by the presence of Sca-1+/CD90+ markers, displayed reduced levels of vimentin and increased expression of genes associated with the Wnt pathway, extracellular matrix remodeling, and testosterone synthesis [15]. The R2 subpopulation, which is mainly characterized by the presence of Sca-1+/CD90-/low cells, exhibited elevated expression of *S*100a4 and Acta2, as well as genes associated with complement activation and cytokine/chemokine pathways. This indicates the presence of myofibroblasts that play a role in tissue repair and inflammation [15]. The R3 subpopulation, characterized by Acta2, but also by *Tagln*, *Mfap2*, and *Mfap4*, was thought to have a probable role in promoting prostate development, the proliferation of blood vessels, and the growth of nerve cells [15]. Crowley et al. discovered separate stromal and immune elements that align somewhat with the stromal subsets described by Kwon et al. [15,17]. To build on the foundational work of these studies in wild-type models, Pakula et al. deepened the understanding of stromal cell diversity of stromal Mes1 and Mes2 [16] and classified tumor-associated stromal cells into eight distinct subpopulations in genetically defined mouse PCa models of prostate cancer [14]. The study revealed that different mesenchymal cell populations were present across GEMMs, with certain populations being specifically linked to oncogenic or tumor suppressor drivers [14] (Figure 2 and Table 1). What remains to be determined is the origin of these expanded subgroups of mesenchymal cells associated with GEMMs. These may arise from pre-existing precursors that expanded during tumor growth or are recruited, transcriptionally induced and clonally expanded, or induced by tumor epithelial cells to adopt molecular and phenotypic characteristics that support their program. Lineage tracing experiments of prostatic mesenchymal populations using Cre lines, currently under development, will provide the necessary insights to address this question.

## 3. Fibroblast Functions and the Transition to Cancer-Associated Fibroblasts (CAFs) in the Prostate Tumor Microenvironment

In healthy conditions, fibroblasts contribute to the normal physiology of the tissue microenvironment. Despite a newly appreciated heterogeneity across different tissues [32], some functions are shared by fibroblasts all over the body. Those include the production of extracellular matrix (ECM) proteins [33], the physical and chemical remodeling of the ECM, and the release of signaling molecules to distinct cellular populations in close proximity within the tissue [34,35,36]. The ECM is an intricate three-dimensional network of macromolecules such as collagens, laminins, glycoproteins, hyaluronans, and proteoglycans. It provides biochemical support to the prostate stroma and serves as an architectural framework for cellular adhesion and migration. Fibroblasts are mainly quiescent during normal prostate development and maintenance and serve as the primary supplier of ECM components. Under physiological conditions, fibroblasts are reversibly activated by external stimuli, like tissue injury and inflammation, to readily respond to the perturbation of the tissue’s steady state [33]. Once the original state is re-established, activated fibroblasts undergo apoptosis in favor of a return to the quiescent phenotype [37]. Similar mechanisms are at the basis of fibroblast activation in cancer—“a wound that does not heal” [38]—with the main difference being that tumor cells continuously remodel the stromal microenvironment to support their survival, growth, and possibly the metastatic phenotype, sustaining a “reactive stroma”. This is characterized by increased collagen deposition [39], a reduction in laminin, and an elevation in tenascin-C expression [40,41]. These activated fibroblasts are commonly referred to as cancer-associated fibroblasts (CAFs). The terms “carcinoma-associated fibroblasts”, “peritumoral fibroblasts”, “cancer-associated myofibroblasts (CAFs)”, “reactive stroma (RS)”, or simply “myofibroblasts” are frequently (and unfortunately) used interchangeably to describe the most abundant cell type in the stroma that surrounds solid malignant tumors [42]. In reality, “CAFs” should be considered as a functional state rather than a separate type of cell. The origins of the many subtypes of CAFs are currently a subject of debate and discord [42,43]. CAFs expressing different surface markers can originate from various cell types [33,44], but can functionally be distinguished in inflammatory CAFs (iCAFs) and myofibroblastic CAFs (myCAFs). The iCAFs are generally thought to promote cancer progression and growth, and they are characterized by an increased expression of inflammatory cytokines and growth factors (e.g., CXCL12, CXCL14, IGF, CCL2) and complement proteins (e.g., C3, C7, CFD). iCAFs appear to be mainly immunosuppressive. CXCL14-expressing iCAFs have been shown to promote the recruitment of monocytes, M2 macrophage polarization, and stimulation of angiogenesis [45]. The protumoral CXCL-14 activity of iCAFs is mediated by overexpression of NOS1 within the iCAFs, resulting in increased intracellular nitric oxide (NO), enhanced oxidative stress, and upregulation of specific transcriptional programs (e.g., SP1, HIF1a) due to NOS1 translocation into the nucleus of iCAFs. The immunosuppressive activity of iCAFs is likely mediated by their interactions with complement pathway proteins. While the role of complement proteins in prostate TME is largely unknown, two studies have reported the presence of iCAFs with transcriptional upregulation of different complement genes in PCa [14,46]. Indeed, we found overexpression of several complement genes (*C3*, *C7*, and *Cfh*) in a cluster of mesenchymal cells (c1 in the 8-cluster partition) represented in all mouse models analyzed. c1 fibroblasts showed a unique set of immunoregulatory and inflammatory genes and members of the interferon-inducible p200 family [14]. Moreover, ligand–receptor analysis revealed interactions between C3 in c1 fibroblasts and integrin receptors in dendritic cells, suggesting a potential role for complement-expressing CAFs as mediators of the stromal–immune crosstalk in the prostate TME [14]. Conversely, myCAFs display an elevated expression of α-SMA and Tagln, and they actively generate and modify the extracellular matrix [47,48]. The contractile characteristics of myCAFs play a role in producing mechanical stresses inside the TME [48]. Interestingly, fibroblasts have the capacity to transform into CAFs via a mechanism referred to as mesenchymal–mesenchymal transition (MMT), resulting in their activation. Exposure to TGFβ1 in dermal fibroblasts increases the quantity of reactive oxygen species (ROS), which initiate a signaling pathway that results in both MMT and the generation of proinvasive signals [49]. In vivo investigations have shown that MSCs are drawn from different organs to prostate tumors through chemotaxis. For instance, CXCL16 binds to CXCR6 receptors present on MSCs, leading to their transdifferentiation into CAFs that, in turn, produce CXCL12, inducing epithelial-to-mesenchymal transition (EMT) [50]. TGF-β1 is also a critical chemical that can effectively attract MSCs to PCa. Human bone marrow-derived embryonic stem cells (hMSCs) exposed to tumor-conditioned medium from breast cancer cells led to increased levels of α-SMA, vimentin, SDF-1, and FSP-1 proteins to activate MSCs. When these were transplanted along with cancer cells into nude mice, they facilitated the development of tumors [51].

## 4. Molecular Mechanisms Underlying CAF Functions

CAFs differ depending on the PCa stage [48,52,53]. On the one hand, the dysregulation and mutation of signaling pathways in epithelial cells have a direct impact on the neighboring stroma, hence playing a crucial role in the initiation and progression of cancer [13,14]. On the other hand, CAFs modulate the gene expression and secretion patterns of PCa cells by providing important signals that promote their growth. These signals trigger a series of signaling events that promote the growth of cancer cells [54] (Figure 3).

AR activation in stromal cells is necessary for the formation of prostate epithelial tumors in genetically modified mice [55]. In 2001, the Cuhna Lab found that hormone therapy after implanting AR-negative epithelial cells into the flanks of castrate mice with AR-positive or AR-negative UGM caused tumors only with AR-positive UGM 55. This study provided evidence that the stroma, and especially stromal AR signaling, promotes cancer. The addition of either wild-type (WT) or androgen receptor-negative mesenchyme (derived from mice with testicular feminization (Tfm)) facilitated the transformation of prostate epithelial cells with repressed PTEN and p53 only when AR was present [56]. In addition, the mating of PTEN+/− mice with ARKO (AR knockout) mice resulted in decreased incidence of spontaneous prostatic intraepithelial neoplasia [57].

The PI3K/AKT/PTEN signaling pathway plays a vital role in controlling CAF activity. Prostate cancer frequently exhibits PI3K hyperactivation, which is caused by loss of PTEN or mutations in PIK3CA/B [58]. We found that deletion of *Pten* in murine prostate mimics human carcinogenesis and causes stromal proliferation around microinvasive prostate intraepithelial neoplasia (mPIN) [59]. Similarly, Wegner et al. (2020) found that PI3K hyperactivation alters the prostate stromal milieu, leading to increased collagen deposition and TGFβ-activated CAFs [60]. The PI3K pathway interacts with several signaling cascades, such as MAPK, WNT, JAK/STAT, EGFR, and Hippo pathways. These interactions collectively affect the activity of CAFs and the growth of tumors [61]. Although there is a significant interaction between these pathways, there is a lack of detailed studies on their functions in controlling CAF activity in prostate cancer.

In PCa, TGFβ is often overproduced by cancer cells [62]. CAFs also have elevated expression of fibroblast growth factors (FGFs), specifically FGF2, FGF7, and FGF10 [63,64,65,66,67]. Akin to TGFβ, FGFs released by CAFs have both autocrine and paracrine actions that stimulate the growth of tumors [63,64,65]. Eliminating stromal FGF2 in the TRAMP mouse model improves survival and hampers tumor development and spread. The FGF-FGFR signaling axis triggers other downstream pathways, including RAS/MAPK, PI3K/AKT/mTOR, and JAK/STAT [68].

The Wnt signaling system is essential for regulating development, organ growth, regeneration, and stem cell differentiation. scRNASeq highlights the elevated activity of Wnt pathways in specific stromal clusters [14]. Wnt ligands (*Wnt5a*, *Wnt2*, *Wnt4*) and receptors (*Fzd1*, *Fzd2*) mediate communication between CAFs and tumor cells [14].

Thus, signals from CAFs operate within a complex web of interactions involving multiple pathways, including PI3K/AKT/PTEN, TGFβ, FGF, and WNT (Table 2). In this context, specific ligand–receptor interactions between stromal and epithelial cells contribute to tumor development, with a prominent role played by Wnt signaling in these processes (Table 3). Additionally, the activation of AR signaling in stromal cells is necessary for the formation of prostate epithelial tumors [14]. In summary, CAFs play a role in modulating diverse and arguably essential biological processes in cancer, including but not limited to the remodeling of the ECM, cancer cell proliferation, invasion, and metastasis.

## 5. CAFs as Modulators of the TME Metabolism

Epithelial tumor cells and stromal cells coevolve and share the same metabolic niche, where both compete for locally available nutrients and secrete different byproducts of their metabolism in the TME. Through soluble paracrine factors, these cells influence each other’s behavior. Lactate, a byproduct of glucose metabolism, is generated during metabolic reprogramming by both epithelial cancer cells and CAFs and plays a critical role in prostate TME [78,79,80]. Prostate CAFs undergo metabolic rewiring, favoring aerobic glycolysis and mitochondrial oxidative stress. In vitro co-culture studies show that activated CAFs increase the expression of the GLUT1 transporter in epithelial cells, enhancing lactate production and its release into the local microenvironment [75]. In turn, PCa epithelial cells in co-culture shift towards aerobic metabolism, increasing lactate uptake using the MCT1 lactate transporter. This acquired metabolic dependence on the lactate of PCa cells is corroborated by reduced survival and growth when MCT1-mediated lactate uptake is inhibited. Activated CAFs also stimulate PCa cells to upregulate oxidative phosphorylation (OXPHOS), which is generally low in normal conditions, through post-translational modification of the pyruvate kinase M2 (PKM2) protein [76]. Epithelial cells utilize lactate, converting it into pyruvate, which fuels OXPHOS. CAF-secreted lactate affects tumor invasion by modulating the immune response, decreasing Th1 effector cells while increasing Tregs, thus promoting tumor progression [81]. This immunomodulatory effect (reduced Th1/increased Tregs) is also seen in tumor explants from PCa patients. Finally, enhanced lactate levels can protect cancer cells from oxidative damage caused by treatments like doxorubicin by upregulating antioxidant enzymes, e.g., SOD1 and SOD2, which reduce free radical species [82]. Citrate derived from the enhanced TCA cycle is utilized in fatty acid synthesis, facilitated by ATP citrate lyase (ACLY), activating de novo lipogenesis, marked by reduced phosphorylation of acetyl-CoA carboxylase (ACC) and increased expression of fatty acid synthase (FASN) [77,83]. Dysregulated lipid metabolism, particularly fatty acid synthesis, has been implicated in tumor progression, recurrence, and chemoresistance [84,85]. Genes involved in lipid metabolism, such as ACLY, ACACA, and FASN, are upregulated in PCA tissues compared with benign epithelium [86]. Upregulation in glutamine metabolism in CAFs supplies carbon and nitrogen for macromolecule synthesis, supporting cellular proliferation and survival [87]. Glutamine deficiency leads to Myc-dependent fibroblast apoptosis and impairs the TCA cycle, enhancing the efficacy of mTOR inhibitors [88]. In docetaxel-resistant PCa cells, a pro-invasive phenotype characterized by epithelial-to-mesenchymal transition (EMT) is associated with increased reliance on OXPHOS over glycolysis (Table 2). Re-expression of miR-205, a microRNA significantly downregulated during EMT and associated with docetaxel resistance, shifts the metabolism back to Warburg-type glycolysis [77]. Prostate-specific deletion of FASN significantly reduces prostate lobe weight and volume, cellular proliferation, microinvasion, and the stromal reaction surrounding intraepithelial neoplasia (PIN) in *Pten* KO mice. This may be a result of stromal reaction to microinvasion or may be a direct effect of inhibition of de novo lipogenesis on mesenchymal cell viability [59]. Diets with increased saturated lipids promote tumor growth, increasing infiltration of CD206+ and PD-L1+ tumor-associated macrophages and FOXP3+ regulatory T cells [89]. These intricate metabolic crosstalks between epithelial tumor cells and CAFs are further modulated by alterations in key regulatory proteins such as p62, which plays a crucial role in shaping the metabolic landscape of the tumor stroma. Prostate tumor stroma with low p62 promotes a protumorigenic interaction between fibroblasts and epithelium, linked to reduced mTORC1 and reprogrammed pentose phosphate and glutathione metabolism, driving an IL6-mediated pro-inflammatory response [90]. p62 loss also activates asparagine metabolism via ATF4, providing nitrogen for tumor proliferation [91]. Lactate conversion to pyruvate reduces p62 by affecting AP-1 transcription factors c-FOS and c-JUN [92]. p62-deficient adipocytes boost osteopontin, promoting tumor fatty acid oxidation and invasion [93]. Taken together, while knowledge of metabolic reprogramming in mesenchymal cells and in the TME, in general, is still scant, these findings suggest that metabolic reprogramming within the TME is a key driver of tumor maintenance and progression.

## 6. Mesenchymal Cells as Biomarkers and Prognostic Indicators

Significant efforts have been made to elucidate the molecular background of the epithelial compartment of PCa to predict its biological and clinical behavior and to define new therapeutic strategies [94,95]. As outlined above, PCa development is associated with substantial alterations in gene and protein expression in epithelial cells, which in turn affect stromal cells and their expression signatures. Molecular alterations in the stroma frequently lead to phenotypic histological changes, commonly referred to as “reactive stroma”, and can be observed as early as in high-grade prostatic intraepithelial neoplasia (HG-PIN) lesions. Ayala et al. [96] were the first to demonstrate that a higher level of PCa reactive stroma is linked to biochemical recurrence (BCR), developing a morphological 4-tier grading system (Reactive Stroma Grade—RSG classification). Since then, other authors confirmed how reactive stroma is associated with the development of castration-resistant prostate cancer (CRPC) [37] and clinical outcomes [67]. Moreover, McKenney et al. [97] showed how incorporating tumor stroma evaluation in Grade Group can improve PCa patients’ prognostic stratification. Analyzing PCa stromal composition with multiplex immunohistochemistry and quantitative image analysis in two retrospective cohorts. Blom et al. [98] confirmed a higher proportion of fibroblasts around high-grade PCa and CRPC compared with low-grade PCa and demonstrated how fibroblasts were the most important cell type in determining PCa patient prognosis.

Several signaling molecules are derived from the mesenchymal compartment of PCa, including Wnt homologs, inflammation mediators, stem cell markers, angiogenesis regulators, tissue growth factors, integrins, and ECM remodeling proteins [99]. One of the most promising of these is Periostin (encoded by *Postn*), an extracellular matrix protein overexpressed in several malignancies, including PCa [100]. In primary tumors of the prostate, its upregulation is associated with high Gleason, shorter BCR, disease-free survival (DFS), and overall survival (OS) [101]. In metastatic castration-resistant PCa (mCRPC), *Postn* is expressed at higher levels than localized tumors. In the SU2C mCRPC cohort, its expression appears to be associated with neuroendocrine molecular features and inversely related to AR expression, perhaps contributing to the progression towards a neuroendocrine phenotype [101]. This was in line with our findings, showing that POSTN is overexpressed in the stroma adjacent to NEPC in both human tumors and mouse models genetically engineered to drive the neuroendocrine phenotype. Additionally, its expression is dramatically increased in human bone metastasis [14].

Using five PCa patient-derived xenograft (PDX) models from different foci of a single original tumor, a 93 gene stromal signature was able to predict metastatic potential in five independent cohorts, including the subpopulation with intermediate Gleason score, potentially improving patients’ management [102]. With a similar approach, Karkampouna et al. [103] used two PCa bone metastasis PDXs—one castration-sensitive and one castration-resistant—and identified stroma-specific transcriptomes and proteomes linked to osteotropism, which were conserved even in non-bone environments (such as mouse subcutaneous sites). Indeed, in the mouse-derived stroma compartment of both PDX, the osteoblastic bone metastasis signature, which included *Postn*, was present [104].

Using laser-capture microdissection (LCM) from human tissue samples to isolate stroma surrounding PCa epithelium, grade 3 reactive stroma (scored according to RSG classification) and matched normal stroma were isolated [39,105]. Up- and down-regulated genes in reactive stroma were related to several pathways, including neurogenesis and metabolic processes. Of note, several growth factors and receptors, including *FGF19*, Wnt signaling (*SFRB1*, *RSPO3*), and *TGFBR*, were upregulated. Genes linked to the immune response, including complement (*C1QA*, *C1QB*, among others), antigen processing (*HLA*), and immune cell activation, were also found to be upregulated in another study [67]. Tyekucheva et al. [13], with the same strategy, isolated and molecularly investigated stromal tissue surrounding stroma of low Gleason and high Gleason scores in radical prostatectomy specimens, identifying a stroma-specific gene signature made by 24 unique differentially expressed genes overexpressed in high Gleason samples, with a higher propensity to metastasize. Again, most of the genes were related to bone remodeling (*BGN*, *COL1A1*, *COL3A1*, *FBLN5*, *LUM*, *SULF1*), potentially explaining PCa osteotropism to immune-related pathways (*C1Q*; *C1QB*, *C1QC*, *C1R*, *C1S*, *HLA-DRB1*), and Wnt signaling (*SFRP2*, *SFRP4*). In validation data, the signature discriminated cases that developed metastasis from those that did not. Of note, this signature comprises all the stromal genes (*BGN*, *COL1A1*, *SFRP4)* of the Oncotype DX Genomic Prostate Score, one of the most used genomic assays to guide PCa management. Using this signature together with the Estimation of Stromal and Immune Cells in Malignant Tumor Tissues Using Expression Data (ESTIMATE) algorithm [106], stromal infiltration from gene expression data was inferred in a large prospective registry cohort (*n* = 5239) and three retrospective institutional cohorts (*n* = 1135). The top decile of stromal infiltration scores was associated with higher CAPRA-S scores, genomic risk scores (Decipher ≥ 0.6), Gleason Score, risk for metastasis, downregulation of DNA repair genes, and radiation sensitivity genomic scores; indeed, postoperative radiation therapy improved metastasis-free survival for this group of patients [107].

Using two long-term follow-up cohorts—the Health Professionals Follow-up Study (HPFS) and Physicians’ Health Study (PHS)—Ma et al. [108] found 92 genes differentially expressed in tumor- and benign-adjacent prostate stromal tissues. These differentially enriched genes were involved in myogenesis, myofibril construction, focal adhesion, and EMT. In particular, 26 genes and 12 pathways involved in stromal remodeling, cholesterol homeostasis, and DNA repair were differently expressed in tumor-adjacent stroma from high- and low-Gleason PCa. In particular, 26 genes and 12 pathways related to stromal remodeling, cholesterol homeostasis, and DNA repair were differentially expressed in the tumor-adjacent stroma of high- and low-Gleason PCa. Additionally, within the benign-adjacent stroma of the two groups, 73 genes and 65 pathways linked to ribosome, translation elongation, and amino acid metabolism were identified. In tumor-adjacent stroma, the expression level of 11 genes associated with TNFα, hypoxia, and hedgehog signaling was linked to PCa-related death.

Finally, another innovative and promising methodology to investigate the prognostic value of the desmoplastic reaction is the computationally-derived image signature [109]. An automated computational approach calculated 242 quantitative metrics of stromal morphology directly on Hematoxylin and Eosin (H&E) stained slides. Nuclear centroids, global and local stromal nuclei connection graphs, nuclear shape and orientation descriptors were utilized to train machine learning and elastic net Cox models to predict BCR. In both independent validation datasets, an African American-specific stromal signature predicted BCR better than clinical standard nomograms (e.g., Kattan and CAPRA-S). Stromal characteristics were substantially related to tumor biomarker expression levels, such as *Rb1*, *Tmrpss2-Erg*, *Pten*, *Erg*, and *c-Myc*.

## 7. Therapeutic Targeting of Mesenchymal Cells in Prostate Cancer

Targeting TME mesenchymal cells for cancer treatment is innovative and promising. These cells can alter tumor progression, angiogenesis, immune infiltrate [110], and therapy resistance, which renders these cells prospective therapeutic targets. Therapeutic approaches aimed at mesenchymal cells include inhibiting signaling pathways, disrupting cell–cell connections, and modulating immunological responses. Some examples are mentioned below.

### 7.1. Inhibition of Key Signaling Pathways

One of the primary strategies for targeting mesenchymal cells involves inhibiting the signaling pathways that drive their tumor-promoting activities. The Transforming Growth Factor-beta (TGF-β) and Wnt/β-catenin pathways and Hedgehog (HH) Signaling Pathway are critical in this regard (Table 4).

#### 7.1.1. Transforming Growth Factor-beta (TGF-β) Pathway

The TGF-β signaling pathway plays a pivotal role in the differentiation of fibroblasts into CAFs. These CAFs contribute to tumor growth and metastasis by secreting ECM components and cytokines, which enhance cancer cell proliferation and migration [62]. TGF-β was shown to induce EMT, mediated by the alternative splicing of CD44, promoting migration and invasion [69,70]. Numerous TGF-β inhibitors are currently under investigation, with galunisertib (LY2157299) being a prominent candidate for PCa treatment [71,72]. This orally administered small molecule selectively inhibits type I TGF-β receptor (TGF-βRI), thereby inhibiting the phosphorylation of SMAD2, a crucial protein in the TGF-β signaling pathway. Galunisertib is presently undergoing a phase II clinical trial (NCT02452008) in combination with enzalutamide [111] to evaluate its efficacy in patients with mCRPC.

#### 7.1.2. Wnt/β-Catenin Pathway

Maintaining mesenchymal cell stemness and proliferation requires the Wnt/β-catenin pathway [131,132]. β-catenin promotes SNAI1, SNAI2, and TWIST expression in different malignancies, affecting both tumor development and metastatic potential [133]. In PCa progression, FGF-receptor-mediated EMT involved SOX9 and Wnt signaling [134]. Chemical inducers of dimerization (CID) caused stepwise progression to EMT-linked cancer in FGFR1 prostate animal models [134]. Two Wnt-targeted therapies are being studied. Inhibiting cancer cell migration and metastasis requires Wnt-5a signaling [112,113]. The Wnt5a mimicking peptide and agonist Foxy-5 limits cancer cell motility and invasion [113]. Foxy-5 was tested for safety and tolerability in metastatic solid tumors, including prostate cancer, in a phase I clinical trial (NCT02020291). Foxy-5, in combination with docetaxel, is being tested in a phase II clinical trial (NCT02655952) in mCRPC patients. Cirmtuzumab, which targets the interaction between ROR1 and Wnt signaling [73,110,114], inhibits Wnt5a-mediated signaling by binding to ROR1 [110], reducing cancer cell proliferation and survival. Cirmtuzumab and docetaxel are also being tested in mCRPC patients in a phase 1B trial (NCT05156905).

#### 7.1.3. Hedgehog Signaling Pathway

The Hedgehog signaling pathway is another crucial pathway implicated in the regulation of mesenchymal cell functions. Aberrant activation of Hedgehog signaling is associated with the promotion of tumor growth and maintenance of cancer stem cell niche. The sonic hedgehog (Shh) inhibitor Vismodegib (GDC-0449) prevents EMT in CRPC cells, resulting in decreased tumor growth [115]. It was tested in locally advanced prostate cancer (NCT01163084) and mCRPC (NCT02115828). These trials, however, did not demonstrate substantial clinical activity [116]. Similarly, sonidegib (LDE225) [117], an inhibitor of the Smoothened (SMO) receptor, evaluated in a separate clinical trial (NCT02111187), showed limited efficacy [73].

### 7.2. Targeting Integrins and Cell Adhesion Molecules

Monoclonal antibodies, aptamers, and various inhibitors of integrins are being used. Abituzumab, a monoclonal antibody that targets αv integrins, inhibits the binding of ligands to αv heterodimers including αvβ1, αvβ3, αvβ5, αvβ6, and αvβ8 [118,119], and has been shown to suppress prostate cancer metastasis by disrupting cell-to-cell and cell-to-extracellular matrix interactions, thereby attenuating invasion and migration potential [118,119]. A phase I clinical trial has demonstrated the activity of Abituzumab in prostate cancer patients with bone metastasis [120]. Unfortunately, the phase II trial showed no significant progression-free survival advantage [121]. Cilengitide, a cyclic pentapeptide and RGD mimetic, antagonizes αvβ3 and αvβ5 integrins with high specificity, leading to reduced tumor cell proliferation and increased apoptosis [122,123]. In vitro studies have confirmed its efficacy in inhibiting osteoclast activity and promoting tumor regression [124]. Although phase I trials indicated clinical activity of Cilengitide as an integrin inhibitor, two phase II trials showed that although well-tolerated, there was no clinical activity in CRCP [122,123].

### 7.3. ECM Degradation

Degradation of the ECM constitutes a pivotal strategy in disrupting the structural integrity and functional support provided by mesenchymal cells within the TME. Matrix metalloproteinases (MMPs), a family of zinc-dependent endopeptidases expressed by stromal cells, play a crucial role in the breakdown of ECM components. Overexpression of MMPs, particularly MMP-2 and MMP-9, has been associated with increased invasiveness and poor prognosis in prostate cancer. Inhibitors of MMPs, such as Batimastat and Marimastat, have demonstrated efficacy in preclinical models by attenuating tumor growth and metastasis through the degradation of ECM components [125,126]. Despite these promising preclinical results, clinical trials with MMP inhibitors have encountered significant challenges. The efficacy observed in preclinical settings has not translated to clinical outcomes, and adverse side effects have been a major concern, highlighting the necessity for more selective and potent inhibitors [127]. For instance, a Phase I/II clinical trial evaluated the efficacy of Marimastat in patients with biochemically relapsed prostate cancer [127]. While a delay in disease progression was observed, evidenced by changes in PSA slope, dose-limiting toxicities hampered its clinical applicability [127].

### 7.4. Immune Modulation

Mesenchymal cells contribute to the immunosuppressive environment within tumors by directly interacting with immune cells and secreting immunomodulatory cytokines. Disrupting these interactions can enhance anti-tumor immune responses. For example, mesenchymal cells often express PD-L1 [135], which binds to PD-1 on T cells, leading to T cell exhaustion and immune evasion by the tumor. Immune checkpoint inhibitors, such as anti-PD-1 and anti-PD-L1 antibodies, can block this interaction, thereby restoring T cell function and promoting anti-tumor immunity [136]. Therefore, clinical trials with immune checkpoint inhibitors, primarily designed to target cancer and immune cells, might also affect MSCs, and combining these inhibitors with therapies targeting MSCs could potentially enhance the overall efficacy of cancer immunotherapy.

Microsatellite instability (MSI) is a key biomarker that has been associated with a subset of prostate cancers and is indicative of defects in DNA mismatch repair (MMR) mechanisms [137]. MSI-high (MSI-H) status is present in a small percentage of prostate cancer cases and has been correlated with increased tumor mutational burden (TMB), which can lead to heightened immunogenicity [129,138,139,140]. This makes MSI-H prostate cancers potential candidates for immune checkpoint inhibitor therapy, such as PD-1/PD-L1 inhibitors. Immune checkpoint inhibitors, which aim to reactivate T-cell responses against tumor cells, have shown promise in MSI-H metastatic prostate cancer, particularly in patients who do not respond to conventional therapies [129]. The application of immune checkpoint blockade in this context has been supported by clinical trials, demonstrating durable responses and prolonged survival in some patients. While not yet a standard approach for all prostate cancer subtypes, the integration of MSI status in the therapeutic decision-making process underscores the growing relevance of immunotherapy in the precision medicine landscape of prostate cancer treatment.

Additionally, targeting other immunosuppressive pathways, such as the CCL2-CCR2 axis, could inhibit the recruitment of monocytes and macrophages that support tumor growth. However, a Phase 2 study of carlumab (CNTO 888), a human monoclonal antibody against (CCL2), in patients with mCRPC was negative [128]. Despite being well-tolerated, carlumab did not effectively block the CCL2/CCR2 axis or demonstrate anti-tumor activity as a single agent in mCRPC. Current clinical trials are now evaluating the combination of CCR2 inhibitors with immune checkpoint inhibitors in cancers other than prostate to overcome the limitations observed with monotherapy.

Beyond pembrolizumab, other immunotherapeutic strategies are being explored for prostate cancer treatment. Ipilimumab, a CTLA-4-targeted antibody, has been studied in metastatic castration-resistant prostate cancer (mCRPC) as monotherapy and in combination with agents like nivolumab, showing modest survival benefits [130]. Sipuleucel-T, an autologous cellular immunotherapy targeting prostatic acid phosphatase (PAP), was among the first FDA-approved treatments for mCRPC, extending overall survival despite limited effects on disease progression [141]. Vaccine therapies [142] like PROSTVAC and GVAX aim to induce strong immune responses, while CAR T-cell therapies targeting PSMA offer promise with high [143] specificity. However, response rates to immunotherapy in prostate cancer remain lower compared with other cancers, likely due to the tumor’s immunosuppressive microenvironment. Research is increasingly focusing on combination therapies with immune checkpoint inhibitors (ICIs) alongside androgen deprivation, radiation, chemotherapy, and novel agents to enhance efficacy and overcome resistance, with ongoing trials seeking to optimize these [144] approaches.

While targeting mesenchymal cells within the PCa TME presents a therapeutic opportunity, the translation of these approaches into effective clinical treatments remains challenging.

## 8. Spatial Techniques to Image the TME

Multiplex imaging techniques are essential for studying the PCa TME, offering detailed insights into the complex interactions between tumor cells, immune cells, and the surrounding stroma. These methods enable the spatial mapping and expression analysis of multiple biomarkers within tumor tissues, helping researchers better understand how the TME influences cancer progression and treatment responses.

Foundational techniques like multiplex immunofluorescence (mIF) and imaging mass cytometry (IMC) allow for the detection of up to 40 protein markers within their spatial context, which is critical for analyzing cellular interactions and disease mechanisms [145]. Cyclic immunofluorescence (CycIF), including methods like COMET (Combinatorial Molecular Evaluation of Tumors), enhances this capability by enabling sequential staining and imaging of up to 60 markers on the same tissue. This allows for more detailed mapping and deeper insights into protein expression and cellular dynamics across the tumor microenvironment [146].

Spatial transcriptomics, such as NanoString’s GeoMX Digital Spatial Profiler (DSP) and Visium, developed by 10× Genomics, goes beyond protein markers by mapping RNA transcripts across tissue sections, providing a detailed view of gene expression while preserving spatial context. These techniques open new possibilities for identifying specific tumor regions that could benefit from targeted therapies [147,148].

Multiplex fluorescence in situ hybridization (mFISH) is another powerful tool for investigating the genetic landscape of the PCa TME. mFISH detects chromosomal alterations like ERG gene fusions and PTEN deletions, which influence both tumor growth and interactions within the microenvironment [149].

Mass spectrometry imaging (MSI) adds a metabolic layer to the analysis, allowing researchers to visualize the spatial distribution of metabolites, lipids, and proteins within the tumor and surrounding tissue. This provides insights into metabolic alterations that may drive disease progression [149].

In conclusion, multiplex imaging techniques provide a comprehensive, multi-dimensional understanding of the PCa TME. By integrating molecular, spatial, and metabolic data, these tools enable researchers and clinicians to better characterize the complex cellular interactions within the TME, leading to more effective and personalized treatment strategies for PCa.

## 9. Conclusions and Future Directions

Cancer-associated fibroblasts (CAFs) are crucial to PCa growth, metastasis, and therapeutic resistance. By engaging with epithelial tumor cells, stromal cells exert control over metabolic reprogramming, immunological evasion, and ECM remodeling, therefore establishing a specialized environment that can facilitate tumor proliferation and aggressiveness. However, the characterization of mesenchymal cells and their signaling pathways requires additional work, and achieving successful results in the clinic continues to be challenging. More work is required to define subpopulations of CAFs and their functional roles at different stages or hormone-dependent and independent states of PCa. It will be essential to create more predictive preclinical models that take into account the TME. Further investigation of mesenchymal cell biology and the development of specific therapeutic approaches targeting this compartment may result in improvements in the treatment of prostate cancer.

## Figures and Tables

**Figure 1 cancers-16-03685-f001:**
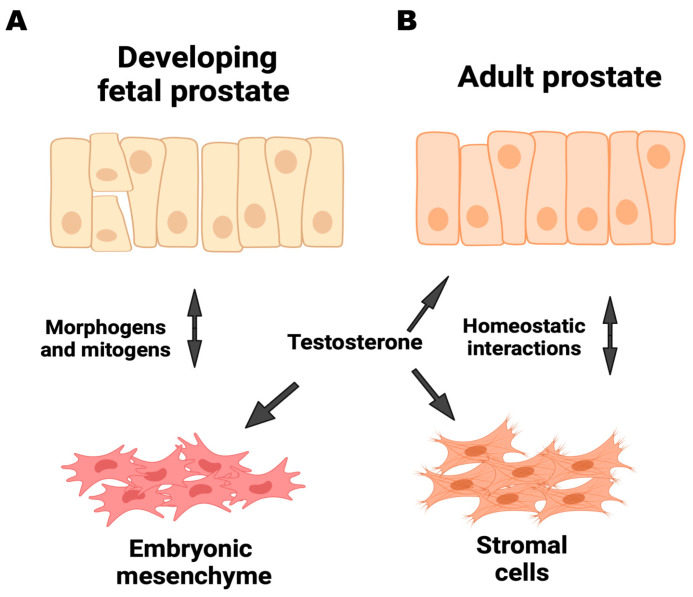
Interaction dynamics between epithelial and stromal/mesenchymal components of the prostate during various stages of development and adulthood. (**A**) Developing fetal prostate: during fetal prostate development, epithelial cells and embryonic mesenchyme engage in reciprocal interactions mediated by morphogens and mitogens. (**B**) Adult prostate: in the adult prostate, stromal cells maintain tissue homeostasis through continuous interactions with the epithelial cells. Testosterone regulates these interactions. Homeostatic feedback mechanisms between the epithelium and stroma are essential for preserving the quiescent state of the prostate tissue. Adapted from [29].

**Figure 2 cancers-16-03685-f002:**
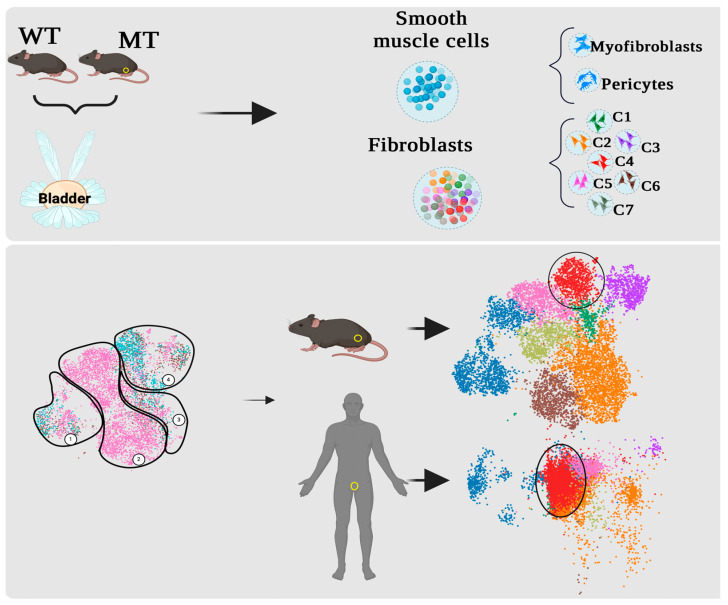
Definition of the stromal microenvironment in murine prostate cancer GEMMs and projections on human tumors. scRNA sequencing studies in wild-type (WT) and mutant (MT) mouse prostates identify mesenchymal cell clusters [14]. **Top Panel**: Among the identified cell types are smooth muscle cells, myofibroblasts, and pericytes. The fibroblast populations are further divided into seven distinct clusters, which overlap with mesenchymal cell clusters M1 and M2, previously described in WT Bl6 mouse [16]. **Bottom Panel**: In mutants, specific mesenchymal clusters are shown, which include myofibroblasts, clusters common to all GEMM models, AR positive and AR negative stromal clusters. Some clusters projected into human prostate cancer are conservation across species [14]. The yellow circle indicates the positions of the prostates in both mouse and human models. The arrows illustrate the distribution of stromal clusters in both species.

**Figure 3 cancers-16-03685-f003:**
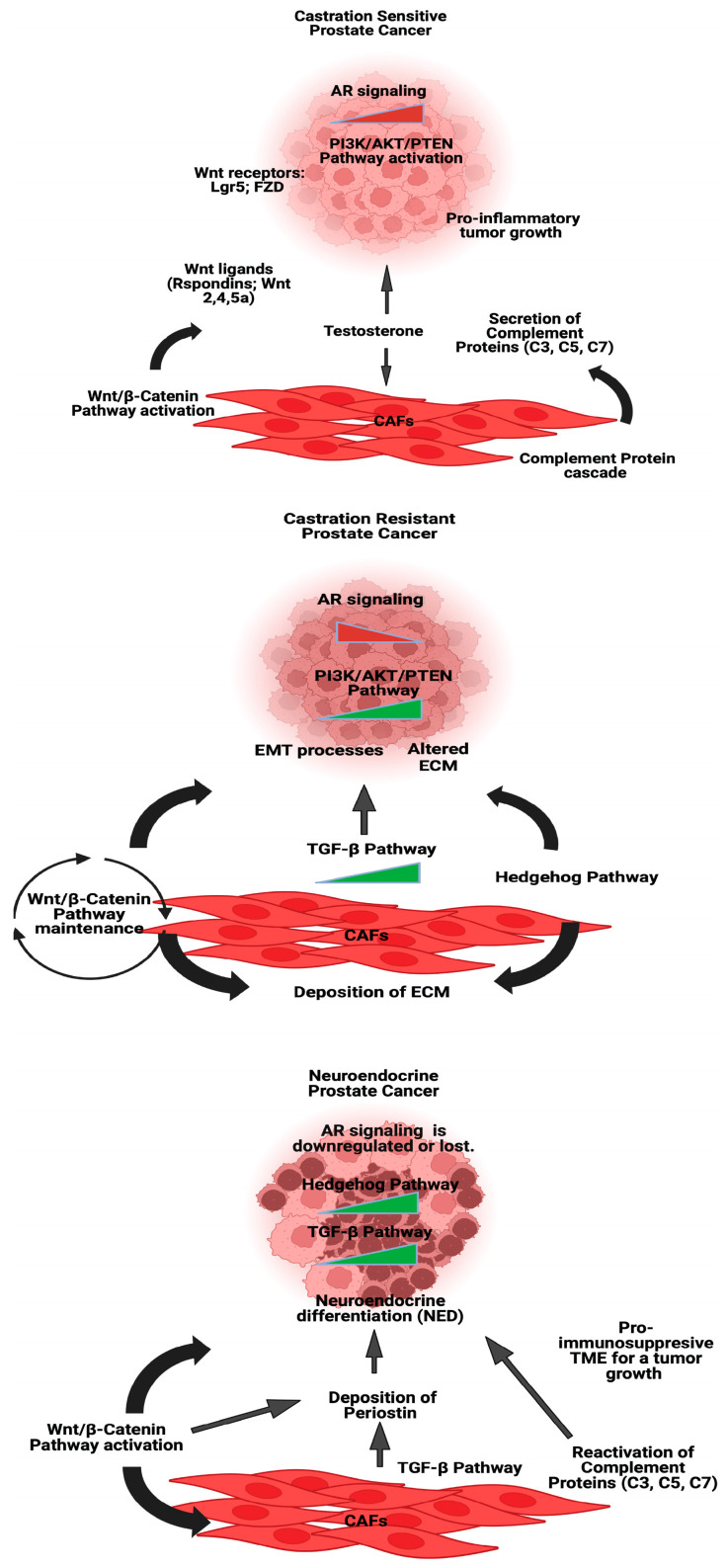
Schematic overview of key signaling pathways in PCa states and the crosstalk with CAFs. This figure illustrates the major signaling pathways involved in three distinct forms of PCa: castration-sensitive prostate cancer (CSPC), castration-resistant prostate cancer (CRPC), neuroendocrine prostate cancer (NEPC), and the interaction between tumor epithelial cells and CAFs.

**Table 1 cancers-16-03685-t001:** Up- and down-regulated genes in castration-sensitive and castration-resistant models of PCa. Table depicts the comparison of top 5 upregulated and downregulated stromal genes in PCa GEMMs and patient specimen, organized into castration-sensitive and castration-resistant categories. The table highlights the genomic similarities and differences between GEMMs (TMPRSS-ERG, HI-MYC, PTEN null) and their human counterparts, with data curated from relevant literature and studies [14,31].

GEMMs for PCa	Top 5 Upregulated Genes in Stromal Cells	Top 5 Downregulated Genes in Stromal Cells	Human PCa	Top 5 Upregulated Genes in Stromal Cells	Top 5 Downregulated Genes in Stromal Cells
Castration-Sensitive Models (*T-ERG*; *HI-MYC*; *Pten−/−)*	*Sult1e1*, *Thbd*, *Ptx3*, *Vit*, *Fgfr4*, *Ar*	*Pdpn*, *Serpinf1*, *Tpm1*, *Mgst3*, *Notch3*	Castration-sensitive cases (6 ERG+ and 3 ERG−)	*BTG2*, *PGC*, *PTX3*, *AR*, *SULT1E1*	*POTEE*, *MYL1*, *SLC6A14*, *TPM1*, *IL22*
Castration-Resistant Models (*Pb-Cre4* ^+*/−*^*;Pten ^f/f^*; *Rb1 ^f/f^;LSL-MYCN ^+/+^*)	*C1q*, *Fn*, *Tnc*, *Loxl3*, *Postn*	*Lgals1*, *Thy1*, *Col13a1*, *Bmp1*, *Gja4*	Castration-resistant cases—bone metastasis by Kfoury et al. 2021 [31]	*BGN*, *LOXL3*, *TNC*, *POSTN*, *C1Q*	*C2*, *SERPINA1D*, *COL5A1*, *BMP1*, *GJA4*

**Table 2 cancers-16-03685-t002:** Summary of key signaling pathways and metabolic rewiring prostate cancer mesenchymal cells. This table delineates the principal signaling pathways, driver genes, and metabolic reprogramming, which are pivotal in the interactions between CAFs and tumor cells. Essential genes implicated in these pathways are enumerated with their functions in mesenchymal cells. These pathways facilitate numerous processes, including fibroblast activation, extracellular matrix remodeling, immunological suppression, preservation of stemness, proliferation of mesenchymal cells, metabolic reprogramming, and tumor formation.

Pathway (References)	Key Genes	Role in the TME or Mesenchymal Cells
TGF-β Signaling [62,69,70,71,72]	TGFBR1, TGFBR2, SMAD2, SMAD3, SMAD4	TGF-β signaling in mesenchymal cells drives fibroblast activation, facilitating ECM remodeling and immune suppression, contributing to tumor progression.
Wnt/β-Catenin Signaling [13,14,15,16,73]	WNT1, CTNNB1 (β-Catenin), RSPO1-3, AXIN2, APC, SFRP2, SFRP4	Wnt signaling maintains stemness and proliferation of mesenchymal cells, contributing to a supportive TME that fosters tumor growth.
Hedgehog Signaling [74]	SHH, PTCH1, SMO, GLI1, GLI2	Hedgehog signaling supports mesenchymal cell proliferation and CAF activation, sustaining the cancer stem cell niche in the TME.
PI3K/AKT Pathway [57,58,59,60]	PIK3CA, PTEN, AKT1, MTOR	Mesenchymal cells activate PI3K/AKT signaling, leading to enhanced tumor cell proliferation, survival, and therapy resistance.
Oxidative Phosphorylation (OXPHOS) [75,76]	PKM2, SOD1, SOD2, COX5B	Mesenchymal cells support oxidative phosphorylation in tumor cells by providing metabolic byproducts like lactate, fueling tumor growth and survival.
Warburg Effect (Aerobic Glycolysis) [77]	GLUT1, MCT1, LDHA, HK2	CAFs in the TME undergo glycolytic reprogramming, producing lactate, which is taken up by tumor cells to fuel their metabolic needs.

**Table 3 cancers-16-03685-t003:** Curated ligand–receptor interactions between stroma and epithelium. This table depicts some of the curated ligand–receptor identified in prostate cancer genetically engineered models 14. It highlights selected interactions between mesenchymal and epithelial components within the tumor microenvironment.

Pathway	Source of Ligand	Target (Receptor)	Ligand	Receptor	pval
WNT-βcatenin	Mesenchyme	Epithelium	Wnt10a	Fzd6-Lrp6	1.91823 × 10^−9^
	Epithelium	Mesenchyme	Wnt4	Fzd2-Lrp6	1.67309 × 10^−6^
ECM-Receptor	Mesenchyme	Epithelium	Col4a6	Itga3-Itgb1	5.43548 × 10^−5^
	Epithelium	Mesenchyme	Lamc2	Itga7-Itgb1	5.88508 × 10^−5^
FGF	Mesenchyme	Epithelium	Fgf2	Fgfr1	1.06555 × 10^−8^
	Epithelium	Mesenchyme	Fgf1	Fgfr1	1.84197 × 10^−7^
Tgfβ	Mesenchyme	Epithelium	Tgfb3	Tgfb3 - (Tgfbr1+Tgfbr2)	1.81695405 × 10^−1^
	Epithelium	Mesenchyme	Tgfb1	Tgfb1 - (Tgfbr1+Tgfbr2)	1.67309 × 10^−6^

**Table 4 cancers-16-03685-t004:** Overview of therapeutic drugs directed at pivotal pathways in CAFs and the TME. The table enumerates different pharmaceuticals, their corresponding classifications, mechanisms of action, and present clinical status. The enumerated pharmaceuticals provide insights into the current endeavors to regulate CAF functioning and metabolic reprogramming inside the TME for cancer treatment.

Drug Name	Class	Target/Mode of Action	Clinical Status
Galunisertib (LY2157299) [70,71,72,111]	TGF-β inhibitor	Selectively inhibits TGF-β receptor I (TGF-βRI), blocking TGF-β signaling, reducing EMT and metastasis.	Clinical Trials (Phase II) in mCRPC (NCT02452008)
Foxy-5 [112,113]	Wnt agonist	Mimics Wnt5a, inhibiting cancer cell migration and metastasis by modulating the Wnt/β-catenin pathway.	Clinical Trials (Phase I/II) for mCRPC (NCT02655952)
Cirmtuzumab [73,108,114]	ROR1 inhibitor	Targets ROR1, blocking Wnt5a-mediated signaling, reducing cancer cell proliferation and survival.	Clinical Trials (Phase I) in mCRPC (NCT05156905)
Vismodegib (GDC-0449) [115,116]	Hedgehog pathway inhibitor	Inhibits Smoothened (SMO) receptor, preventing EMT and reducing tumor growth in CRPC.	Clinical Trials (NCT01163084) in mCRPC
Sonidegib (LDE225) [117]	Hedgehog pathway inhibitor	Inhibits SMO receptor, targeting the Hedgehog signaling pathway to reduce cancer stem cell maintenance.	Clinical Trials (NCT02111187) in mCRPC
Abituzumab [118,119,120,121]	Integrin inhibitor	Targets αv integrins, disrupting cell-to-cell and cell-to-ECM interactions, inhibiting metastasis.	Phase I Clinical Trial (bone metastasis)
Cilengitide [122,123,124]	Integrin inhibitor	Antagonizes αvβ3 and αvβ5 integrins, inhibiting osteoclast activity, reducing tumor growth and invasion.	Phase I/II Clinical Trials (no significant efficacy in CRPC)
Marimastat [125,126,127]	MMP inhibitor	Inhibits MMPs (especially MMP-2 and MMP-9), reducing ECM degradation, tumor growth, and metastasis.	Phase I/II Clinical Trials (limited efficacy and dose-limiting toxicities)
Carlumab (CNTO 888) [128]	CCR2 inhibitor	Monoclonal antibody against CCL2, inhibiting recruitment of tumor-promoting monocytes and macrophages.	Phase II (failed to show efficacy in mCRPC)
Pembrolizumab [129]	PD-1 inhibitor	Immune checkpoint inhibitor that blocks PD-1 on T cells, restoring immune function against cancer cells.	Marketed (being tested in combination with other agents)
Nivolumab [130]	PD-1 inhibitor	Inhibits PD-1, enhancing immune response against tumor cells.	Ongoing clinical trials
Ipilimumab [130]	CTLA-4-targeted antibody	Inhibits CTLA-4, sending negative signals to T cells.	Phase III clinical trial (failed to reach overall survival benefit)

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
