# Peer review of "Deciphering the Tumor Microenvironment in Prostate Cancer: A Focus on the Stromal Component"

_cancers, 2024, doi:10.3390/cancers16213685_

Round 1

Reviewer 1 Report

Comments and Suggestions for Authors

This reviewer tries to decipher the tumor microenvironment in prostate cancer. They focus on cellular interactions and therapeutic targets. It examines how CAFs interact with epithelial tumor cells, examine how these promote tumor growth and regulate immune responses via signaling pathways. They summarized that mesenchymal cells are biomarkers and prognostic indicators, and stromal composition and activation predict clinical outcomes. Also, they provided how targeting TME mesenchymal cells can improve cancer therapy. It is helpful for researchers in this field. Some major comments listed below.

Major comments:

1. Too much emphasis on background knowledge and insufficient description of the latest developments throughout the article. Please remove any known background knowledge and only keep necessary statements, without citing too many earlier literatures. There are already many such reviews in this field. So, this review can only help researchers if it can summarize the latest developments and propose new perspectives.

2. The application of charts in the review can facilitate readers' reading. Please increase the display of charts as much as possible. The beauty of charts is also an important indicator of the level of review.

3. The overall citation and listing of literature still give readers the feeling of piling up literature. It would be even better if the author could provide some summary and forward-looking viewpoints based on existing literature.

4. To join the author's research group for related work, including published or unpublished works.

5. The length is too long, and there are too many references cited. This is not convenient for readers to read.

Author Response

Reviewer #1:

This reviewer tries to decipher the tumor microenvironment in prostate cancer. They focus on cellular interactions and therapeutic targets. It examines how CAFs interact with epithelial tumor cells, examine how these promote tumor growth and regulate immune responses via signaling pathways. They summarized that mesenchymal cells are biomarkers and prognostic indicators, and stromal composition and activation predict clinical outcomes. Also, they provided how targeting TME mesenchymal cells can improve cancer therapy. It is helpful for researchers in this field. Some major comments listed below.

Major comments:

  1. Too much emphasis on background knowledge and insufficient description of the latest developments throughout the article. Please remove any known background knowledge and only keep necessary statements, without citing too many earlier literatures. There are already many such reviews in this field. So, this review can only help researchers if it can summarize the latest developments and propose new perspectives.

Response: Thank you for your feedback regarding the balance between background knowledge and the latest developments. To this end, we have removed older or redundant literature, keeping only strictly necessary citations that provide context for newer findings. We have also shortened the background section. We believe that the approach we have taken presents the most up-to-date and detailed advancements in the field of prostate cancer TME, focused on the stromal component since many reviews already address the immune microenvironment. We specifically highlight, compare and contrast key recent studies on this topic, including Karthaus et al., Crowley et al. 2020, Kwon et al. 2019, and our own work, Pakula et al. 2024, which represent the latest and most detailed descriptions of the prostate cancer tumor microenvironment. We summarize significant new insights into mesenchymal cell diversity, stromal-epithelial interactions, and their impact on tumor progression and therapeutic resistance. Furthermore, we discuss how emerging insights into the stromal microenvironment could translate into new therapeutic strategies. The new perspective are focused on specific stromal changes in murine models that accompany some of the most common genetic alterations. These contribute to tumor initiation and progression. Importantly, they are conserved in patient samples, providing novel prognostic and predictive biomarkers and potential therapeutic targets.

  1. The application of charts in the review can facilitate readers' reading. Please increase the display of charts as much as possible. The beauty of charts is also an important indicator of the level of review.

Response: Thank you for your recommendation concerning the incorporation of charts in the review. The incorporation of visual representations significantly improves the clarity and comprehensibility of intricate material. We have therefore added more figures and tables that show important connections and pathways between the prostate tumor microenvironment (TME) and the neoplastic epithelial component. In particular, a figure has been added that shows the main signaling pathways active in castration-sensitive, castration-resistant and neuroendocrine prostate tumors (Figure 3 page 7). We added 4 detailed tables: Table 1 (page 4) showing up-and down-regulated genes in castration-sensitive, castration-resistant and neuroendocrine prostate tumors, Table 2 (page 10) summarizing key signaling pathways in the TME and mesenchymal cells; Table 3 (page 11) depicting curated Ligand-Receptor interactions between stroma and epithelium in the TME of GEMMs for PCa; and Table 4 (page 16)  listing therapeutic agents (both approved and in clinical trials) along with putative mechanism of action.

  1. The overall citation and listing of literature still give readers the feeling of piling up literature. It would be even better if the author could provide some summary and forward-looking viewpoints based on existing literature.

Response:

In response, we have thoroughly reviewed the manuscript and made adjustments to ensure that the literature cited is directly relevant to the key points discussed. Specifically, on pages 3, 4, 6, 9, 10, 12-16, and 17, we condensed the presentation of literature to focus on the most pivotal studies while adding more synthesis of the findings rather than simply listing them. For example, in the discussion of the role of mesenchymal cells in the prostate TME (pages 3-4), we emphasized key papers such as Pakula et al. (2024), Karthaus et al. (2020), and others to reflect the most current understanding of TME and stromal interactions. In the chapter titled CAFs as modulators of the TME metabolism (page 9), we elaborated on the latest data on dysregulated lipid metabolism by the Loda laboratory and highlighted the recent intricate metabolic crosstalk between epithelial tumor cells and CAFs, which is further modulated by alterations in key regulatory proteins such as p62 (from the Moscat and Diaz-Meco laboratories). Additionally, between pages 12 and 17, we included forward-looking perspectives that discuss potential future research directions and clinical implications based on existing literature. These include the need for further exploration into stromal subpopulations, metabolic reprogramming, and the development of therapeutic approaches targeting the TME in prostate cancer. We believe these modifications will better serve the readers by providing a balanced view of past findings while guiding attention toward areas with potential for future breakthroughs. We hope these changes, along with specific page references, adequately address the reviewer's concerns

  1. To join the author's research group for related work, including published or unpublished works. 

Response:

We appreciate the reviewer's suggestion to incorporate more details from our research group, including both published and unpublished work. In the revised manuscript, we have highlighted our relevant published work, particularly Pakula et al. 2024, where we explore key interactions in the prostate tumor microenvironment. Additionally, we referenced Kfoury et al. 2021, where stromal clusters were projected onto bone metastasis data, further expanding our understanding of stromal-epithelial interactions in advanced prostate cancer. Unpublished data from our group are being actively developed, with a focus on stromal subpopulations, mesenchymal alterations, and their therapeutic implications in prostate cancer progression. We are currently working on projects that delve into the role of stromal reprogramming in treatment-resistant prostate cancer, which will further expand on the concepts discussed in this review. We believe that by integrating these findings with existing literature, our manuscript provides a comprehensive perspective on both the current state and future directions of research into prostate cancer stromal dynamics. 

  1. The length is too long, and there are too many references cited. This is not convenient for readers to read.

Response: As mentioned above, we have shortened the text considerably and reduced the number of references as requested.

Reviewer 2 Report

Comments and Suggestions for Authors

The authors have reviewed the prostate cancer tumor microenvironment, with specific emphasis on cellular interactions and therapeutic targets, particularly cancer-associated fibroblasts (CAFs), that might be playing a dual role in prostate cancer progression, either facilitating or inhibiting tumor growth depending on the stage. Following points should be addressed to improve the article and make it comprehensive.

1.       Abstract is not well written and needs to be improved.

2.       Typographical errors should be corrected for instance subheading 7.1.1. is missing.

3.       Gene name abbreviations have been mostly italicized but not for all (example line 404), make it uniform.

4.       A new figure showing the major interactions and possible links between all the main pathways will impart a holistic understanding of PCa.

5.       A table enlisting the major involved canonical pathways and the key genes should be included. Also, a table for therapeutic drugs (in market and under clinical trials) for PCa and their primary mode of action should be added. 

Comments on the Quality of English Language

Needs language improvements.

Author Response

The authors have reviewed the prostate cancer tumor microenvironment, with specific emphasis on cellular interactions and therapeutic targets, particularly cancer-associated fibroblasts (CAFs), that might be playing a dual role in prostate cancer progression, either facilitating or inhibiting tumor growth depending on the stage. Following points should be addressed to improve the article and make it comprehensive.

  1. Abstract is not well written and needs to be improved.
    Response: Thank you for pointing this out. The re-wrote the abstract which now gives a more succinct and focused summary that better reflects the content of the review. This specifically focuses on the role of mesenchymal cells in prostate cancer initiation and progression. The updated abstract also emphasizes the therapeutic relevance of the stromal microenvironment. We have now included a Simple Summary before the Abstract, written in layman's terms. We have eliminated abbreviations from the abstract.

  1. Typographical errors should be corrected for instance subheading 7.1.1. is missing.

Response: We have fixed typographical errors and the erroneous numbering of the subheadings.

  1. Gene name abbreviations have been mostly italicized but not for all (example line 404), make it uniform.

Response:  This has been corrected.

  1. A new figure showing the major interactions and possible links between all the main pathways will impart a holistic understanding of PCa.

Response: Thank you for the suggestion. We have now added a new figure 3 that offers a comprehensive overview of the principal pathways and interactions between stromal cells and tumor epithelium. The figure is not meant to be exhaustive and comprehensive but merely points out what emerges in the literature and key interactions pointed out in several studies.

  1. A table enlisting the major involved canonical pathways and the key genes should be included. Also, a table for therapeutic drugs (in market and under clinical trials) for PCa and their primary mode of action should be added. 

Response: As requested, we have now added 4 detailed tables to the updated manuscript: Table 1 (page 4) showing up-and down-regulated genes in castration-sensitive, castration-resistant and neuroendocrine prostate tumors, Table 2 (page 10) summarizing key signaling pathways in the TME and mesenchymal cells; Table 3 (page 11) depicting curated Ligand-Receptor interactions between stroma and epithelium in the TME of GEMMs for PCa; and Table 4 (page 16)  listing therapeutic agents (both approved and in clinical trials) along with putative mechanism of action.

Reviewer 3 Report

Comments and Suggestions for Authors

excellent review on prostate cancer TME.  The one minor English error can be corrected on editing after acceptance.

Author Response

Excellent review on prostate cancer TME.  The one minor English error can be corrected on editing after acceptance.

Response: We thank this reviewer for your positive feedback.

Reviewer 4 Report

Comments and Suggestions for Authors

The authors focus on the relationship between epithelial and stromal cells in prostate cancer, with a specific emphasis on genetic alterations that play an important role in inducing mesenchymal alterations in the TME.

Overall, the manuscript is well-written with detailed materials. Although there are certain areas where the manuscript can be improved to enhance clarity and impact. Particularly, prostate specific cell-cell interactions should be highlighted, to less extent general considerations of histology independent interactions in tumor tissues. The paper should point out differences between hormone sensitive tumors and those castration resistant tumors or neuroendocrine differentiated ones.

The whole concept of the review should be put in context with perspective experimental, diagnostic and therapeutic concepts to attract the readers’ interest.

In detail:

An overview about genetic aberrations in prostate cancer, discussed aberrations and specific consequences on cell-cell-interactions should be given e.g., in a table.

Separation of oncogene addictions (targets) and non-oncogene addictions (targets) is important, just in prostate cancer.

Please give an overview (Table) about biomarkers indicating stromal activity and corresponding genetic link in mouse models, human tissue. It should be indicated whether data from human tissue are derived from retrospective/prospective studies and the respective endpoints should be mentioned.

There is a long paragraph about CAFs. What are the prostate cancer-specific qualities of CAFs on the respective genetic background in prostate cancer cells?

No prostate cancer specific summary on CAFs has been given: ‘In summary, CAFs play a role in diverse and arguably essential biological processes in cancer, including but not limited to the remodeling of the ECM, cancer cell proliferation, invasion and metastasis. These effects are mediated by important signaling pathways such as TGFβ, PI3K/AKT, FGF and Wnt. Ligand-receptor interactions enable communication within the tumor microenvironment.’

‘Taken together, these findings suggest that metabolic reprogramming in the TME is closely associated with tumor maintenance and progression.’ This is a general remark, not prostate cancer specific! May be prostate cancer growth inhibited by metabolic reprogramming?

‘A comprehensive understanding of their mechanisms, interaction with cancer cells and contribution to the prostate cancer TME will provide insight into future therapeutic treatments which may alter the cancer metastatic TME to a cancer-eliminating one which may benefit metastatic prostate cancer patients.’ The so called ‘future treatments’ are standards in clinical prostate cancer subgroups!! DOI: 10.3389/fimmu.2022.1001297 , DOI: 10.1158/1078-0432.CCR-23-3403

The paper has a special paragraph on immune cells in prostate cancer. The immune microenvironment landscape and immune-related subtypes in prostate cancer is important to discuss. Why did you not mention MSI and immune checkpoint inhibitor therapy in prostate cancer? 

In context with the pharmacologic modeling of Wnt signaling, the overview of Vallee doi: 10.3389/fimmu.2018.00745 is important, also in context with prostate cancer.

Future directions:

Imaging mass cytometry (IMC) is an emerging imaging technology that exploits the multiplexed analysis capabilities of the CyTOF mass cytometer to make spatially resolved measurements for tissue sections.

Particularly in prostate cancer the use of non-oncogene addiction targets in therapy is routine: Therefore, simultaneous targeting of tumor and cancer cells is already established by androgen deprivation therapy and now combined with novel targeted therapies.

Hallmarks of prostate cancer may be targeted. Such strategies are also targeting cellular interactions between tumor and stroma cells. E.g., Wnt signaling, prostate cancer metabolism etc. may be pharmacologically modulated also in prostate cancer doi: 10.3389/fimmu.2018.00745.

The review points out that targeting cellular interactions or single cell compartments might be successful. Thereby, such interactions are commonly considered like oncogenic events, aberrant signaling – specific target - response, as you exemplified. Only in rare cases, you may observe this sequence: Microsatellite instability is prerequisite for the successful use of immune checkpoint inhibitors in prostate cancer, but generally multi-targeting approaches of cellular interactions in tumor tissues are leading to therapeutic success.

Unlocking immunity in prostate cancer: e.g., doi: 10.1038/s41585-023-00739-w, ‘vaccines as treatments for prostate cancer’, immune checkpoint inhibitors etc.

Author Response

The authors focus on the relationship between epithelial and stromal cells in prostate cancer, with a specific emphasis on genetic alterations that play an important role in inducing mesenchymal alterations in the TME. 

Overall, the manuscript is well-written with detailed materials. Although there are certain areas where the manuscript can be improved to enhance clarity and impact. Particularly, prostate specific cell-cell interactions should be highlighted, to less extent general considerations of histology independent interactions in tumor tissues. The paper should point out differences between hormone sensitive tumors and those castration resistant tumors or neuroendocrine differentiated ones. The whole concept of the review should be put in context with perspective experimental, diagnostic and therapeutic concepts to attract the readers’ interest.

Response: Thank you for your insightful comments. In response to your suggestions, we have made several key updates to the manuscript to enhance clarity and impact, particularly regarding cell-cell interactions and the distinction between hormone-sensitive, castration-resistant, and neuroendocrine differentiated tumors.

To address the point about signaling pathways and cell-cell interactions in prostate cancer TME (Tables 1 - page 4, 2- page 10 and 3- page 11; and Table 4- page 16), we have provided curated ligand-receptor interactions based on our recent work in Pakula et al. 2024. This includes a detailed analysis of interactions between epithelial and stromal cells that are highly relevant to the TME in human prostate cancer. This limited and non exhaustive list of curated genes and interactions provide what we interpret as the most important ones affecting prostate cancer behavior.

Furthermore, we recognize the importance of distinguishing between hormone-sensitive and castration-resistant tumors. Our earlier work was performed on four genetically engineered mouse models (GEMMs), none of which were castrated. Three models are hormone sensitive (spell them out) and a fourth naturally evolves into a neuroendocrine cancer was a low/negative AR expressor in both epithelium and stroma, and can thus be considered castration-resistant. In the same paper, the nine human cases we analyzed by scRNASeq were all hormone-naïve. By comparison, the bone metastases data from Kfoury et al. 2021 Cancer Cell focused on castration-resistant cases. In light of this, we have created a simple table (Table 1) in the manuscript highlighting the differentially expressed stromal genes (upregulated and downregulated) between castration-sensitive models (T-ERG, PTEN-/-, and Hi-MYC in the mouse, as well as our nine human cases) and castration-resistant models (PRN in the mouse and Kfoury's data in humans). These genes partially define the stromal clusters defined in the review.

Finally, we have ensured that this review underscores future experimental, diagnostic, and therapeutic implications that derive from the meticulous dissection of the microenvironment, in this case, focusing on the stromal one. We believe the updated manuscript now provides a more focused and relevant overview of the current understanding of the prostate cancer stromal microenvironment.

In detail:

An overview about genetic aberrations in prostate cancer, discussed aberrations and specific consequences on cell-cell-interactions should be given e.g., in a table.

Separation of oncogene addictions (targets) and non-oncogene addictions (targets) is important, just in prostate cancer.

Please give an overview (Table) about biomarkers indicating stromal activity and corresponding genetic link in mouse models, human tissue. It should be indicated whether data from human tissue are derived from retrospective/prospective studies and the respective endpoints should be mentioned.

…”There is a long paragraph about CAFs. What are the prostate cancer-specific qualities of CAFs on the respective genetic background in prostate cancer cells? 

No prostate cancer specific summary on CAFs has been given: ‘In summary, CAFs play a role in diverse and arguably essential biological processes in cancer, including but not limited to the remodeling of the ECM, cancer cell proliferation, invasion and metastasis. These effects are mediated by important signaling pathways such as TGFβ, PI3K/AKT, FGF and Wnt. Ligand-receptor interactions enable communication within the tumor microenvironment.’”…

Response: Thank you for your insightful comments. In response to your suggestions, we have created four comprehensive tables to address the specific points raised:

Table 1 (page 4): Up-and Down-regulated Genes in Castration-Sensitive and Castration-Resistant Models of Prostate Cancer – This table highlights genetic aberrations across castration-sensitive (e.g., TMPRSS2-ERG, HI-MYC, PTEN null) and castration-resistant/neuroendocrine (e.g., Pb-Cre4 +/-;Pten f/f; Rb1 f/f;LSL-MYCN +/+- PRN model) models, comparing selected upregulated and downregulated genes in stromal cells shared between GEMMs and human prostate cancer. This includes oncogene (e.g., MYC, ERG) and non-oncogene addictions (e.g., PTEN) – if we interpret “oncogene and non-oncogene addictions” correctly - and their specific impact on stromal-epithelial interactions.

Table 2 (page 10): Summary of key signaling pathways and metabolic rewiring prostate cancer mesenchymal cells. – This table summarizes some of the most upregulated signaling pathways involved in stromal-epithelial interactions within the prostate cancer microenvironment, including TGF-β, Wnt/β-catenin, Hedgehog, and PI3K/AKT. We have also outlined the genes linked to these pathways, separating pathways driving mesenchymal activation, extracellular matrix (ECM) remodeling, and metabolic reprogramming in both mouse and human models.

Table 3 (page 11) : Curated Ligand-Receptor Interactions Between Stroma and Epithelium– This table depicts some of the curated ligand-receptor identified in prostate cancer genetically engineered models. It highlights selected interactions between mesenchymal and epithelial components within the tumor microenvironment.

However, it is important to emphasize that many of these genetic aberrations and stromal biomarkers have only been projected onto human models based on experimental data from mouse studies. Ongoing research, including work from our group and others, is actively seeking to define and validate these findings in human tissues.

Table 4 (page 16) Overview of therapeutic drugs directed at pivotal pathways in cancer-associated fibroblasts (CAFs) and the tumor microenvironment (TME). The table enumerates different pharmaceuticals, their corresponding classifications, mechanisms of action, and present clinical status. The enumerated pharmaceuticals provide insights into the current endeavors to regulate CAF functioning and metabolic reprogramming inside the tumor microenvironment for cancer treatment.

‘Taken together, these findings suggest that metabolic reprogramming in the TME is closely associated with tumor maintenance and progression.’ This is a general remark, not prostate cancer specific! May be prostate cancer growth inhibited by metabolic reprogramming?

Response: Thank you for pointing this out. We have now restricted our comments to metabolic reprogramming in prostate stromal cells. In prostate cancer, a lot is known about changes in metabolic reprogramming in the epithelium but studies focused on the microenvironment are scant and not exhaustive. Here, we attempted to summarize some of the data present in the literature and speculate on potential stromal epithelial interaction in metabolic state. Again, data are limited to date.

‘A comprehensive understanding of their mechanisms, interaction with cancer cells and contribution to the prostate cancer TME will provide insight into future therapeutic treatments which may alter the cancer metastatic TME to a cancer-eliminating one which may benefit metastatic prostate cancer patients.’ The so called ‘future treatments’ are standards in clinical prostate cancer subgroups!! DOI: 10.3389/fimmu.2022.1001297 , DOI: 10.1158/1078-0432.CCR-23-3403 

The paper has a special paragraph on immune cells in prostate cancer. The immune microenvironment landscape and immune-related subtypes in prostate cancer is important to discuss. Why did you not mention MSI and immune checkpoint inhibitor therapy in prostate cancer? 

“Hallmarks of prostate cancer may be targeted. Such strategies are also targeting cellular interactions between tumor and stroma cells. E.g., Wnt signaling, prostate cancer metabolism etc. may be pharmacologically modulated also in prostate cancer doi: 10.3389/fimmu.2018.00745.

The review points out that targeting cellular interactions or single cell compartments might be successful. Thereby, such interactions are commonly considered like oncogenic events, aberrant signaling – specific target - response, as you exemplified. Only in rare cases, you may observe this sequence: Microsatellite instability is prerequisite for the successful use of immune checkpoint inhibitors in prostate cancer, but generally multi-targeting approaches of cellular interactions in tumor tissues are leading to therapeutic success.

Unlocking immunity in prostate cancer: e.g., doi: 10.1038/s41585-023-00739-w, ‘vaccines as treatments for prostate cancer’, immune checkpoint inhibitors etc.”

Response: Thank you for the thoughtful feedback regarding the scope and content of the review, particularly concerning the discussion of immune-related subtypes and microsatellite instability. Immunotherapy, including immune checkpoint inhibitors, is of course standard treatment for specific subgroups of prostate cancer patients, particularly those with mismatch repair-deficient tumors. While this review focuses on the stromal microenvironment and does not want to duplicate or expand upon  excellent reviews focused on the immune microenvironment, we revised text to some context in this regard. For example, a new paragraph has been added to discuss MSI and its role in immune checkpoint inhibitor therapy in prostate cancer, referencing key clinical trials. The section on future directions has also been expanded to emphasize emerging immunotherapies beyond ICIs, such as new vaccines, CAR-T cells, and bispecific antibodies. (page 13: Ma et al 2022, page 15: Xiong Z, et al. 2024; Patsoukis N et al. 2020; Nava Rodrigues D et al. 2018. Le DT et al. 2018).).

In context with the pharmacologic modeling of Wnt signaling, the overview of Vallee doi: 10.3389/fimmu.2018.00745 is important, also in context with prostate cancer.

Response: Thank you for your suggestion. We have incorporated relevant insights from this review into the discussion (page 14, Vallee A et al. 2018).

Future directions: 

Imaging mass cytometry (IMC) is an emerging imaging technology that exploits the multiplexed analysis capabilities of the CyTOF mass cytometer to make spatially resolved measurements for tissue sections.

Particularly in prostate cancer the use of non-oncogene addiction targets in therapy is routine: Therefore, simultaneous targeting of tumor and cancer cells is already established by androgen deprivation therapy and now combined with novel targeted therapies.

Response: Thank you for the comment. We expanded on this by integrating information about multiplex imaging techniques, which offer detailed insights into the spatial dynamics of tumor microenvironment interactions. (page 17, Rivest F, et al 2023; Hernandez S, et al.2022; Williams CG et al. 2022; Chen KH et al 2015 ).

Round 2

Reviewer 2 Report

Comments and Suggestions for Authors

Authors have addressed all the questions raised previously. Addition of new figure and tables along with content improvement has improved the article substantially. Few typographical corrections are still needed though.

Reviewer 4 Report

Comments and Suggestions for Authors

Thank you for improving the manuscript